# Assessing the validity of the zero-velocity update method for sprinting speeds

**Gerard Aristizábal Pla**[1], **Douglas N. Martini**[1], **Michael V. Potter**[2],
**Wouter Hoogkamer**[1]*

1 Department of Kinesiology, University of Massachusetts, Amherst, MA, United States of America,
2 Department of Physics and Engineering, Francis Marion University, Florence, SC, United States of America

* whoogkamer@umass.edu

**Data Availability Statement:** Our full data set is available on OSF: https://osf.io/q6ku3/

**Funding:** The authors received no specific funding for this work.

## Abstract

The zero-velocity update (ZUPT) method has become a popular approach to estimate foot kinematics from foot worn inertial measurement units (IMUs) during walking and running. However, the accuracy of the ZUPT method for stride parameters at sprinting speeds remains unknown, specifically when using sensors with characteristics well suited for sprinting (i.e., high accelerometer and gyroscope ranges and sampling rates). Seventeen participants performed 80-meter track sprints while wearing a Blue Trident IMeasureU IMU. Two cameras, at 20 and 70 meters from the start, were used to validate the ZUPT method on a stride-by-stride and on a cumulative distance basis. In particular, the validity of the ZUPT method was assessed for: (1) estimating a single stride length attained near the end of an 80m sprint (i.e., stride at 70m); (2) estimating cumulative distance from $\sim 20$ to $\sim 70$ m; and (3) estimating total distance traveled for an 80-meter track sprint. Individual stride length errors at the 70-meter mark were within -6% to 3%, with a bias of -0.27%. Cumulative distance errors were within -4 to 2%, with biases ranging from -0.85 to -1.22%. The results of this study demonstrate the ZUPT method provides accurate estimates of stride length and cumulative distance traveled for sprinting speeds.

## Introduction

Sprinting performance is key in track and field [1] and team sports [2, 3]. Athletes and coaches looking to improve sprinting performance can assess different sprinting performance determinants, such as stride frequency, stride length and speed during different phases of a sprint and use this information to evaluate races and training runs. Stride length estimates are particularly useful in hurdle and jump events where take off position needs to be optimal. Stride length and running speed estimates can be extracted with a variety of commercially available devices such as (high speed) video cameras [4], radar guns [5], laser beams [6], smartphone apps [4], optical measurement systems [7], and resistance devices [8]. Inertial measurement units (IMUs) also have the potential to assess sprint performance. An IMU generally consists of tri-axial accelerometers, angular rate gyroscopes and magnetometers [9]. Primary benefits of IMUs are affordability, easy set-up, minimal interference with the runner's performance, and

**Competing interests:** The authors have declared that no competing interests exist.

the ability to provide instantaneous feedback. One of the challenges of IMUs is that position measures are sensitive to integration drift [10, 11]. Integration drift error can be minimized with the zero-velocity update (ZUPT) method.

The IMU-based ZUPT method can provide estimates of foot kinematics with shoe-worn IMUs in real-life settings [12–18]. The ZUPT method works in a four-step process. In the first step, the stride segmentation, raw IMU signals are used to detect zero velocity instants when the foot is (approximately) stationary on the ground. In the second step, the rotational orientation estimation, IMU orientation in space is estimated using an extended Kalman filter. In the third step, the translational velocity estimation, linear accelerations are integrated between two successive zero velocity instants. Exploiting the fact that the foot is at zero velocity at the start and end of each stride, a linear drift correction is performed to correct the velocity back to zero during each zero-velocity instant, thus reducing the integration drift errors. In the fourth step, the trajectory formation, stride parameters (e.g., stride length) are obtained by integrating foot velocities.

The implementation of the ZUPT method for sprinting has two main challenges: issues related to the hardware and issues related to the inertial data processing. The inertial data processing for sprinting is challenging because the foot may not have a clear zero-velocity period, as opposed to level walking [13, 16–18], which could affect the correction of integration drift errors. The hardware can become an issue because peak accelerations and angular velocities during sprinting might be outside the measurement range of commercially available IMUs and sample frequencies might be insufficient to capture highly dynamic movements, potentially further impacting the external validity of stride lengths obtained with the ZUPT method.

The traditional ZUPT method [10] has been shown to yield accurate estimates at running speeds up to 6.4m/s [12, 19–21]. Bailey and Harle [19] investigated the accuracy of the traditional ZUPT method for speeds up to 3.4m/s on a treadmill and obtained biases of 0.002 ±0.029m and 0.03±0.02m/s for the estimation of foot clearance and mean step velocity, respectively. Brahms et al. [21] investigated the accuracy of the traditional ZUPT method for speeds up to 4.36m/s during overground running and obtained a bias of -0.032±0.150m for the estimation of stride length. Bailey and Harle [20] and Potter et al. [12] additionally investigated the effects of several IMU specifications. Bailey and Harle [20] investigated the effect of accelerometer range and sampling frequency, while Potter et al. [12] investigated the effect of accelerometer range, gyroscope range and sampling frequency. The study by Potter et al. [12] was conducted in outdoor environments and the speeds analyzed were much higher (up to 6.4m/s) than those analyzed by Bailey and Harle [20] (up to 3.4m/s). Therefore, the ZUPT estimates obtained by Potter et al. [12] were more impacted by limitations in the IMU hardware (i.e., sensor characteristics). Quantifying stride metrics with the traditional ZUPT method at speeds up to 6.4 m/s over 100 meters showed that sampling frequency significantly impacts the traditional ZUPT estimates of individual stride parameters [12]. In addition, saturation due to low gyroscope range (i.e., 750˚/s) or accelerometer range (i.e., 24g) also reduced stride estimate accuracy [12]. Though, the authors noted that acceptable estimates (i.e., errors below 5%) may still be obtained if the amount of data that is lost due to saturation remains small (i.e., 1.5% in acceleration signals and 2.6% in angular velocity signals) [12].

De Ruiter et al. [14] assessed the validity of the ZUPT method for sprinting speeds (i.e., speeds over 8m/s), using an IMU with a ±16g and a ±2000˚/s range, sampling at 500Hz. To mitigate the effects of accelerometer saturation, the ZUPT method was modified with respect to the traditional ZUPT method [14]. De Ruiter et al. [14] defined a stride by the time interval between two consecutive initial contacts rather than between two consecutive zero-velocity times. Rather than applying a linear drift correction, integration drift error in velocity was corrected by first identifying the minimum values (i.e., velocity offsets) in the filtered velocity

signals between 20 and 100 milliseconds following initial contact, then the velocity offsets were subtracted from the raw velocity signals and all data points prior to the samples with minimum values were imposed to be zero [14]. Finally, offset corrected velocities were integrated to obtain stride length. For peak sprint speeds of 8.42±0.85m/s, the obtained bias and limits of agreement were reasonably accurate (i.e., -2.51±8.54%) [14]. Based on the findings by Potter et al. [10], the sprint speed results reported by De Ruiter et al. [14] could have been affected by the IMU hardware specifications, especially their relatively low accelerometer range (±16g). This suggests that the accuracy of the traditional ZUPT method on a stride-by-stride basis for track sprinting speeds could be improved when using sensors with characteristics better suited for sprinting (i.e., higher ranges and sampling rates).

Therefore, the aim of this study was to assess the validity of the traditional ZUPT method for: (1) estimating a single stride attained near the end of an 80m sprint (i.e., stride at 70m); (2) estimating cumulative distance from ∼20 to ∼70 m; and (3) estimating total distance traveled for a 80-meter track sprint. These estimations were performed using data collected with an IMU with a high accelerometer range (±200g) and sampling frequency (1125Hz). Based on Potter et al. [12], we hypothesized that IMU-based stride lengths and cumulative distances would be within 5% from stride estimates obtained with a camera-based capture system.

## Methods

### Participants

Seventeen participants over 18 years old were enrolled in this study (4 women, 24.6±6.1yrs, 1.77±0.09m, 71.8±10.3kg, all mean±SD; recruitment period: March 14, 2022 –May 25, 2022). Inclusion criteria were ability to run 7m/s or faster and being free of injury for at least three months prior their testing session. Exclusion criteria included any orthopedic, cardiovascular, or neuromuscular conditions that would affect sprint performance. Each participant provided a written informed consent approved by the University of Massachusetts Amherst Institutional Review Board (#3143).

### Experimental protocol

We collected IMU data using the commercially available Blue Trident IMeasureU IMU (see Table 1 for sensor characteristics). We placed the IMU on the right shoe; specifically, to the medial dorsal aspect of the foot [12]. The IMU was attached using double-sided tape and strapped down with Hypafix tape to reduce motion artefacts.

We placed two cameras (Apple iPhone 12, 1080 pixels at 240Hz), mounted on tripods to capture footfalls at 20m and 70m to establish the exact distance traveled from the start to each footfall closest to each of these marks. Following de Ruiter et al. [14], we placed the iPhone cameras 8 meters away from the track lane that subjects were running in. We choose to place the cameras at 20 and 70m to ensure participants were running near top speed (at 70m) and to separate the acceleration phase from the maximal speed phase (at 20m). The camera at the 70m mark captured the distance between two consecutive footfalls for the right foot to validate the traditional IMU-based ZUPT method for a single stride. We placed tape marks at set distances throughout the full capture space to calibrate the iPhone camera views and to account for triangulation in the analysis (see below). The distance between consecutive tape marks was seven meters. We used an additional mobile camera at 40m to count the number of strides taken until the participants ran through the view of the static cameras. This enabled us to compare the same stride from the IMU and each stationary iPhone camera.

Each subject completed a self-selected warm-up, then ran an 80-meter sprint at maximal effort on an outdoor track. We instructed subjects to stand still for ∼15 seconds before the

**Table 1. Blue Trident IMeasureU IMU sensor specifications.**

| | |
|---|---|
| Accelerometer sample frequency (Hz) | 1125, 1600 |
| Accelerometer range (g) | ± 16, ± 200 |
| Accelerometer bit-resolution (bits) | 16, 13 |
| Accelerometer resolution ($m/s^2$) | 0.0048, 0.48 |
| Accelerometer bandwidth (Hz) | 1125, 1600 |
| Accelerometer anti-aliasing filter cut-off frequency (Hz) | 473, 800 |
| Accelerometer noise density ($mg/\sqrt{Hz}$) | 0.23 |
| Gyroscope sample frequency (Hz) | 1125 |
| Gyroscope range (deg/s) | ± 2000 |
| Gyroscope bit-resolution (bits) | 16 |
| Gyroscope resolution (deg/s) | 0.061 |
| Gyroscope bandwidth (Hz) | 773.5 |
| Gyroscope anti-aliasing filter cut-off frequency (Hz) | 473 |
| Gyroscope noise density ($deg/s/\sqrt{Hz}$) | 0.015 |

Left values correspond to the low-g accelerometer. Right values correspond to the high-g accelerometer.

start of the sprint with the vertical projection of the IMU aligned with the start line. This was done to subtract the gyroscope fixed bias [22].

## IMU analysis

The raw IMU signals were downloaded and analyzed with customized software in Python (Python Software Foundation, Delaware, DE, USA). The customized software used the high-g accelerometer (±200g) whenever the low-g accelerometer (±16g) saturated. Firstly, the 1600Hz high-g accelerometer signals were linearly down sampled to match the 1125Hz low-g accelerometer signals. Then, a cross-correlation analysis was run to find any phase shift between the two signals. Finally, zero padding was used to make sure the maximum value of the cross correlation occurred at zero-lag. We included this step because the resolution of the high-g accelerometer is much lower than the resolution of the low-g accelerometer. Thus, it would not be possible to obtain accurate IMU-derived stride lengths by just using the high-g accelerometer. Then, we calculated stride lengths using the ZUPT method, introduced above and described in detail in Potter et al. [12].

The detection of stationary periods was done using customized software in Python (Python Software Foundation, Delaware, DE, USA). First, initial contacts were detected as maximum peaks in the resultant acceleration signal. Following Skog et al. [16], thresholds were then adjusted for the gyroscope and acceleration signals to identify 8 samples with the lowest magnitude within the first 25% of the time between initial contacts. These 8 samples represent 5 percent of an average contact time of 150ms that the foot would be stationary. The stationary period was defined as the single sample with the minimum angular velocity during the longest consecutive series of identified samples.

## Validation

Individual stride lengths were added to yield the estimated total distance traveled ($D_{calc}$). We calculated the cumulative distance error ($D_{err}$), for 0-∼20, ∼20-∼70 and 0-∼70

meters, as follows [10]:

$$D_{err} = \left( \frac{D_{calc}}{D_{truth}} - 1 \right) * 100\%, \tag{1}$$

where $D_{truth}$ is the real distance traveled (i.e., $\sim 20$, $\sim 50$ and $\sim 70$ meters), obtained from the camera recordings. Total known distances traveled were divided by the duration extracted from the IMU (i.e., the time between two consecutive stationary periods) to obtain average speeds.

Individual IMU-based stride lengths errors ($S_{err}$) were calculated for the stride nearest to the 70-meter mark as follows:

$$S_{err} = \left( \frac{S_{calc}}{S_{truth}} - 1 \right) * 100\%, \tag{2}$$

where $S_{truth}$ is the known stride length obtained from the camera and $S_{calc}$ is the stride length obtained with the IMU-based ZUPT method. The stride length was divided by the IMU-derived stride duration (i.e., the time between two consecutive stationary periods) to obtain stride speed.

$S_{truth}$ and $D_{truth}$ were calculated by tracking the IMU's position using Kinovea software (www.kinovea.org). We used known points (i.e., tape marks) and the known distances between those points to define a perspective grid. That was done to obtain a calibrated camera view that allowed for accurate calculation of stride lengths and the exact total distance traveled.

For $S_{truth}$, we identified video frames at or near midstance for two consecutive right steps. Then we placed a marker on top of the IMU and drew the vertical projection of the IMU to the ground. We were then able to calculate the horizontal distance between the two vertical projections to obtain stride length. For $D_{truth}$, we calculated the horizontal distance between the vertical projection of the IMU to the tape mark, denoting 20 or 70 meters from the start.

Three trained researchers independently analyzed all the videos with Kinovea. If the difference between the results obtained from the three researchers for the same trial was smaller than 2%, then the average of the three was taken. If the difference was larger than 2%, we removed the results from the researcher whose result was further away and the average of the two remaining researchers was taken. If the average of the two still did not lead to a difference smaller than 2%, then all researchers reanalyzed the same trial and the process was repeated.

## Statistical analysis

We present all results in the text as mean values ± SD. We used Bland-Altman analysis to determine the agreement between the individual stride and cumulative distances obtained with Kinovea and with the IMU. Limits of agreement are defined as bias ± 1.96xSD (i.e., 95% confidence interval). We used simple and multiple linear regressions to investigate the effect of body mass and speed on the individual stride and cumulative distance errors. For the simple linear regressions, errors were included as dependent variables and the speeds were independent variables. For the multiple linear regressions, errors were included as dependent variables and the speeds and masses as independent variables. The relative magnitude of the effects of different variables were quantified with the standardized betas (β), with β<0.29 being a small effect, 0.30<β<0.49 being a medium effect, β>0.50 being a large effect. Alpha level was set *a priori* to 0.05 for the slope of the regression.

## Results

Descriptive characteristics of all participants are presented in Table 2. Participants reached stride speeds ranging from 6.0 to 9.3m/s (8.00±0.88m/s) and stride lengths from 3.45 to 4.73m (4.07±0.30m). We found that the individual stride length errors (Fig 1A) were between -6 and 3% for the individual stride at 70m, with a bias (± limits of agreement) of -0.27±4.61% (Fig 2A) and distance errors from -0.24 to 0.11m with a bias of -0.01±0.19m. Participants reached 20-meter distances ranging from 20.25 to 23.44m (22.23±0.99m) with average speeds ranging from 4.5 to 6.3m/s (5.51±0.51m/s). We found that 20-meter cumulative distance errors (Fig 1B) were smaller, ranging from -4 to 2%, with a bias of -0.85±3.47% (Fig 2B) and distance errors from -0.82 to 0.30m and a bias of -0.19±0.76m. Participants ran 20-70-meter ranging from 47.62 to 56.24m (51.32±2.84m) with average speeds ranging from 6.7 to 9.5m/s (8.29 ±0.78m/s), while 20–70-meter cumulative distance errors (Fig 1C) remained within -4 to 2%, with a bias of -1.22±3.67% (Fig 2C) and distance errors from -2.06 to 0.83m with a bias of -0.62±1.89m. Participants ran the full 70-meter sprints ranging from 70.39 to 79.24m (73.55 ±2.55m) with average speeds ranging from 6.0 to 8.2m/s (7.19±0.65m/s), with 70-meter cumulative distance errors (Fig 1D) within -4 to 2%, with a bias of -1.11±3.50% (Fig 2D) and distance errors from -2.72 to 1.05m and a bias of -0.81±2.57m. Note that average speeds for 20-70m (Fig 1C) were higher than those for the single stride at 70m (Figs 1A and 3). Individual and multiple linear regression for the stride and cumulative distance errors are presented in Table 3. Speed had a large significant negative effect and body mass had a small significant positive effect on the individual IMU-based stride length error, and on the 20-70-meter cumulative distance error. Speed had a large significant negative effect on the 70-meter cumulative distance error.

## Discussion

The aim of this study was to evaluate the accuracy of the IMU-based ZUPT method on a stride-by-stride basis and on total distance traveled for track sprinting. We found that

**Table 2. Descriptive characteristics for all participants.**

| ID number | Mass (kg) | Height (m) | Shoe type | IMU estimated maximum speed 80m sprint |
|---|---|---|---|---|
| 1 | 70.8 | 1.86 | Trainers | 7.34 |
| 2 | 81.7 | 1.88 | Trainers | 7.69 |
| 3 | 65.8 | 1.78 | Spikes | 7.85 |
| 4 | 54.4 | 1.65 | Trainers | 6.55 |
| 5 | 72.6 | 1.65 | Trainers | 7.45 |
| 6 | 83.9 | 1.88 | Spikes | 8.70 |
| 7 | 81.7 | 1.85 | Trainers | 8.50 |
| 8 | 78.0 | 1.78 | Trainers | 7.44 |
| 9 | 77.1 | 1.78 | Trainers | 8.26 |
| 10 | 56.7 | 1.68 | Trainers | 6.25 |
| 11 | 86.2 | 1.88 | Trainers | 8.18 |
| 12 | 74.8 | 1.68 | Trainers | 8.10 |
| 13 | 81.6 | 1.83 | Spikes | 8.38 |
| 14 | 67.1 | 1.70 | Spikes | 9.20 |
| 15 | 59.0 | 1.70 | Spikes | 8.40 |
| 16 | 54.4 | 1.63 | Spikes | 8.54 |
| 17 | 74.8 | 1.91 | Spikes | 9.02 |

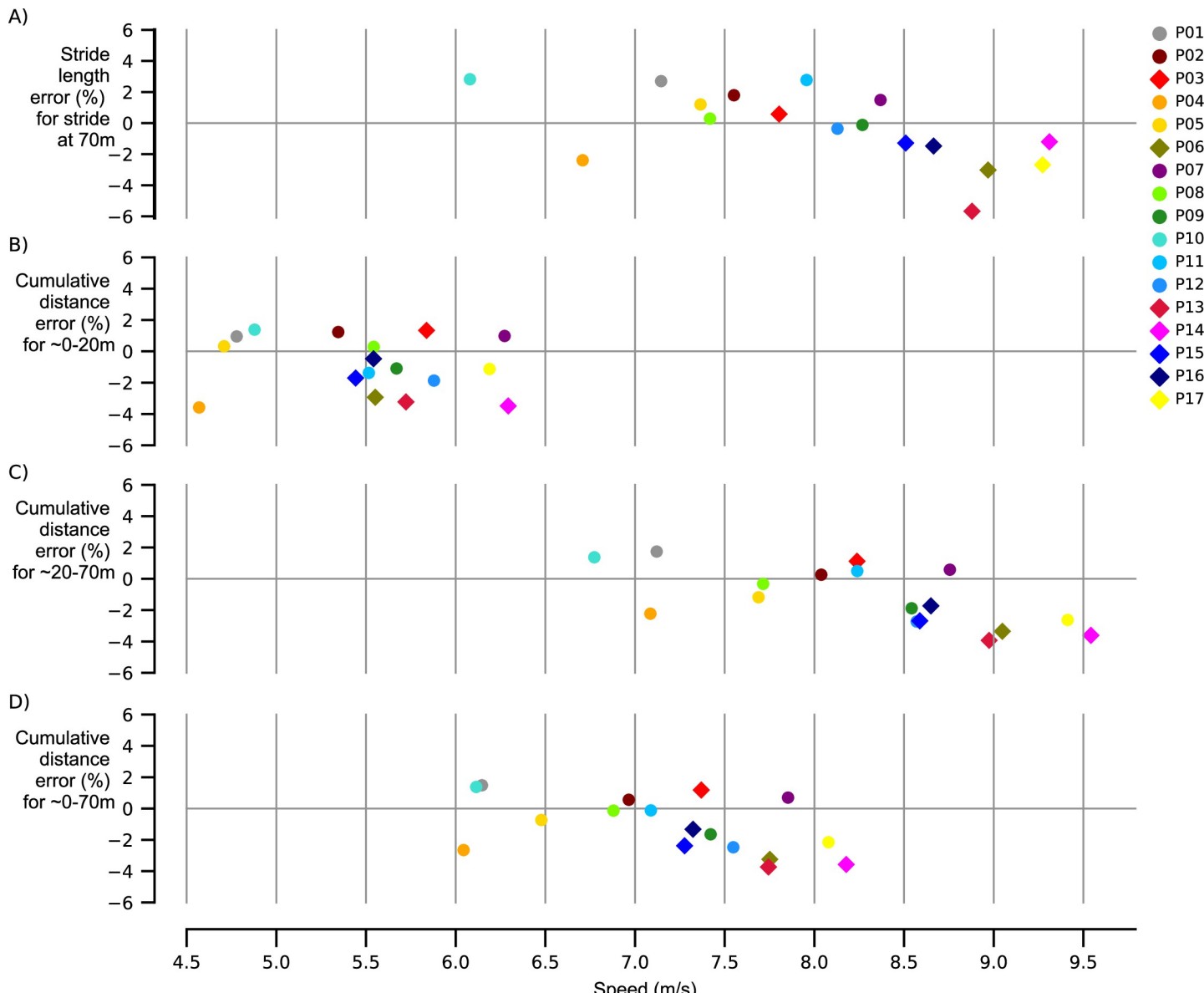

**Fig 1. Stride length error and cumulative distances errors versus average speed comparing IMU to Kinovea.** Stride length errors were between -6 and 4% across speeds, while cumulative distance errors were between -4 and 2% across speeds. Colors represent different subjects. Circles represent participants wearing trainers and diamonds represent participants wearing spikes.

individual stride and cumulative distance errors were low (i.e., within -6 to 3% and -4 to 2%, respectively).

For peak sprinting speeds of 8.00±0.88m/s, we obtained an average bias ± limits of agreement of-0.27±4.61% (Fig 2) or -0.01±0.19m. Our findings are similar to those obtained for running at slower speeds (a bias of -0.032 ± 0.150m for overground running up to 4.36 m/s [21], but better than those obtained for sprinting (a bias of -2.51±8.54% for peak sprint speeds of 8.42±0.85 m/s [14]). To mitigate accelerometer saturation, the de Ruiter et al. [14] provided an alternative approach of the traditional ZUPT method [10]. However, their results were likely still affected by accelerometer saturation, especially during ground contact, as suggested

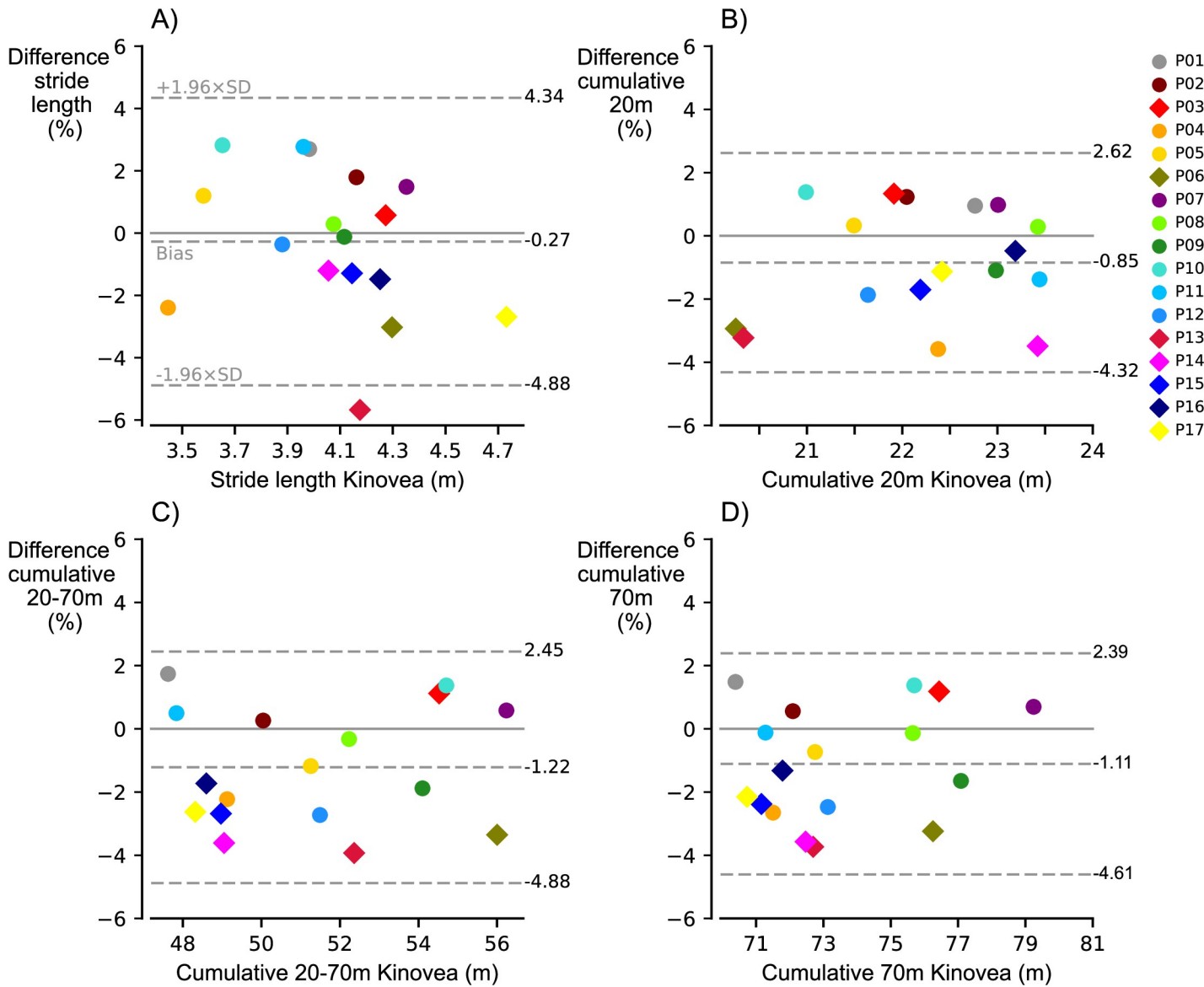

**Fig 2. Comparison of IMU-derived measures against Kinovea-derived measures. A** For the individual stride length at 70m the average bias (dashed line black) ± limits of agreement (dashed line grey) were -0.27±4.61%. **B** For cumulative distance over the first 20m the average bias ± limits of agreement were -0.85±3.47%. **C** For cumulative distance from 20 to 70m the average bias ± limits of agreement were -1.22±3.67%. **D** For cumulative distance over the full 70m the average bias ± limits of agreement were -1.11±3.50%. Limits of agreement are defined as bias ± 1.96xSD (i.e., 95%).

by evidence that accelerometer saturation leads to a degradation in accuracy of the ZUPT method [12]. The present study suggests that without the presence of accelerometer saturation, the traditional ZUPT method provides accurate and superior estimates for sprinting speeds over the modified ZUPT method.

Without accelerometer saturation, we obtained stride length errors within -6 to 3% for stride speeds up to 9.3m/s. Potter et al. found that for 100-meter average speeds of 6.5m/s, a ±16g accelerometer led to a 15% underestimation of a 100-meter distance, compared to a 100g accelerometer [12]. De Ruiter et al. [14] obtained stride length errors as high as 30% for stride speeds of 9m/s using a ±16g range IMU. Our results indicate that the accuracy of the ZUPT method on a stride-by-stride basis for high sprinting speeds can be improved when using

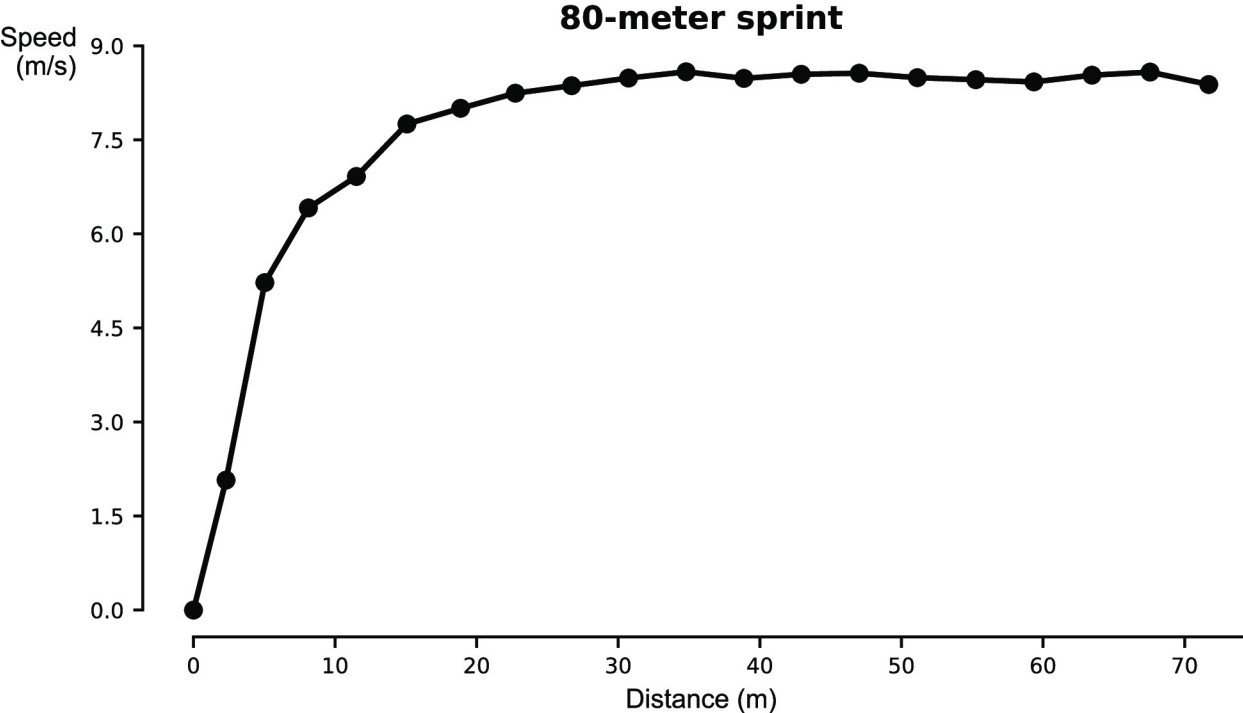

**Fig 3. IMU-derived speed for an 80-meter sprint on a stride-by-stride basis as a function of distance traveled for a representative subject (P09).** Circles represent individual stride speeds.

sensors with specifications better suited for sprinting velocities (i.e., higher accelerometer ranges). In addition, we observed 20-70-meter cumulative distance errors within -4 to 2%, demonstrating that the ZUPT method yields acceptable results (i.e., errors of 5% or less) for stride speeds as high as 9.5m/s.

One of the main potential sources of error of the ZUPT method could have been gyroscope saturation during ground contact. Similar to Potter et al. [12], our findings suggest that gyroscope saturation leads to underestimated stride lengths and cumulative distance errors. We observed that underestimation increased with running speed and for participants wearing spikes. Participants wearing spikes exhibited gyroscope saturation in more steps than participants wearing trainers, which could lead to a larger percent of gyroscope data lost due to saturation. This saturation was mostly exhibited at ground contact when the foot was in plantarflexion. Trainers have thicker midsole foams than spikes [23], which provide more

**Table 3. Simple and multiple linear regression results.** Significant differences are highlighted in bold.

| Error | Linear regression type | *p* value | Adjusted R² value | Standardized β speed (% per m/s) | Standardized β mass (% per kg) |
|---|---|---|---|---|---|
| Stride 70m | Simple | **0.010** | 0.32 | -0.60 | |
| | Multiple | **0.020** | 0.34 | -0.68 | 0.26 |
| Cumulative 20m | Simple | 0.524 | -0.04 | -0.17 | |
| | Multiple | 0.792 | -0.01 | -0.20 | 0.08 |
| Cumulative 20-70m | Simple | **0.004** | 0.39 | -0.65 | |
| | Multiple | **0.009** | 0.41 | -0.74 | 0.26 |
| Cumulative 70 | Simple | **0.037** | 0.21 | -0.51 | |
| | Multiple | 0.079 | 0.18 | -0.60 | 0.23 |

cushioning, likely acting as a physical low pass filter that reduces peak magnitudes in kinematics. For sprinting in spikes with less cushioning than trainers, higher peak angular velocities can be expected, which could lead to gyroscope saturation at high sprinting speeds. Potter et al. [12] found that the cumulative distance errors remained below 5% when the percentage of gyroscope data lost due to saturation was below 2.6% [12]. This suggests that the amount of data that we lost from saturation was low, even though we could not quantify what percentage of data we lost due to saturation.

For the participants that did not exhibit gyroscope saturation (n = 6), we obtained cumulative and individual distance errors within -3 to 3% for speeds up to 8.5m/s. Those errors are very similar to error reported for level walking [10]. Though we did not observe any signal saturation for these participants, that does not mean that there was no saturation, as theoretically signal saturation could be camouflaged by IMU internal preprocessing. However, the internal preprocessing of the Blue Trident IMeasureU consists of a digital low band pass filter with a cut-off frequency of 473Hz, while the sensor bandwidth is 773.5Hz. Those cut-off frequencies are well beyond what can be expected during human sprinting.

While the bias for stride length that we obtained was only -0.27%, the 95% limits of agreement were ±4.6%. In elite sports, 1% differences in performance could be the difference between a gold medal and no medal. Without the presence of IMU saturation, the accuracy of estimated stride lengths during sprinting could be improved and can be of practical relevance in training applications. Stride length estimates are particularly useful in hurdle and jump events where take off position needs to be optimal. Further, accurate stride length estimations allow for accurate speed calculation. From the sprint speed curve presented in Fig 3, sprinting performance determinants such as maximal speed [1] can be extracted. Such a speed curve could also be supplemented with the method proposed by Samozino et al. [24] to obtain estimates of force outputs (e.g., ratio of force). Additionally, the estimates of sensor displacement and orientation used in the ZUPT method can be used to obtain other stride parameters that could be levered in sprint training settings (e.g., contact time, swing time, step frequency, plantar- and dorsiflexion angles). In addition to track and field sports [1], sprinting performance is key in team sports [2, 3]. Thus, this method could provide coaches with key determinants to assess and improve sports performance. Note that this method only requires a single IMU attached to the athlete's foot, the IMU does not need to be aligned with anatomical axes and outcomes are minimally affected by subject-specific characteristics (i.e., body mass). Note that the significant effect of body mass (Table 3) could have been confounded by other factors.

Future research should try to assess the validity of the ZUPT method for high sprinting speeds (i.e., speeds over 9 m/s) as our results were affected by speed, but with sensors that have specifications even better suited for sprinting. Such sensors would not admit gyroscope saturation (>±2400°/s) and have good resolutions (>16bits) and appropriate bandwidths (>500 Hz) and sample frequencies (>1000 Hz) to capture high frequency impacts during ground contact. In commercially available IMUs there are often trade-offs between sensor range, resolution and sampling frequency and we are unaware of any commercial devices that possess all these recommended characteristics. Our data suggests that accurate measures can be obtained with IMUs with specifications similar to the IMU used here (Table 1), but the significant effect of speed also suggests that for even faster speeds stride length is likely to be increasingly under estimated with the current methods and specifications. Future research should directly compare the effects of spikes and trainers on the accuracy of the ZUPT method, as our interpretation on the effects of spikes could have been confounded by the fact that spikes were worn for different subjects. Our findings suggest that with the appropriate sensors, the ZUPT method could possibly be used to compare the performance of different track spikes [23] as well as to test elite sprinters.

In conclusion, the results of this study demonstrate the accuracy of the ZUPT method for sprinting speeds, even with the presence of gyroscope saturation. Cumulative and individual distance errors remained within -6 to 3% for speeds ranging from 6m/s to 9.5m/s.

## Acknowledgments

We thank UMILL lab members who helped with data collections, Shane Schwartz and Herlandt Lino for their help with the Kinovea analyses, Dr. Alex Shorter and Dr. Loubna Baroudi for sharing their insights and Dr. Leia Stirling for her help with software development and ZUPT implementation.

## Author Contributions

**Conceptualization:** Gerard Aristizábal Pla, Wouter Hoogkamer.

**Formal analysis:** Gerard Aristizábal Pla.

**Investigation:** Gerard Aristizábal Pla.

**Methodology:** Douglas N. Martini, Michael V. Potter, Wouter Hoogkamer.

**Software:** Gerard Aristizábal Pla, Michael V. Potter.

**Supervision:** Wouter Hoogkamer.

**Visualization:** Gerard Aristizábal Pla.

**Writing – original draft:** Gerard Aristizábal Pla, Wouter Hoogkamer.

**Writing – review & editing:** Gerard Aristizábal Pla, Douglas N. Martini, Michael V. Potter, Wouter Hoogkamer.

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
