## [Decision Letter · Decision Letter 0]

7 Aug 2023

PONE-D-23-19191Assessing the validity of the zero-velocity update method for sprinting speedsPLOS ONE

Dear Dr. Hoogkamer,

Thank you for submitting your manuscript to PLOS ONE. After careful consideration, we feel that it has merit but does not fully meet PLOS ONE’s publication criteria as it currently stands. Therefore, we invite you to submit a revised version of the manuscript that addresses the points raised during the review process.

ACADEMIC EDITOR:Both Reviewers recognized the effort made by the Authors in prodicing methodological update with respect to ZUPT approach. Moreove,r the quality of the manuscript is overall good. However, some major concerns have to be addressed in order to render the manuscript suitable for publication.

We look forward to receiving your revised manuscript.

Kind regards,

Andrea Tigrini, Ph.D.

Academic Editor

PLOS ONE

3. We note that Figure 1 in your submission contain copyrighted images. All PLOS content is published under the Creative Commons Attribution License (CC BY 4.0), which means that the manuscript, images, and Supporting Information files will be freely available online, and any third party is permitted to access, download, copy, distribute, and use these materials in any way, even commercially, with proper attribution. For more information, see our copyright guidelines: http://journals.plos.org/plosone/s/licenses-and-copyright.

Additional Editor Comments:

Both Reviewers recognized the effort made by the Authors in prodicing methodological update with respect to ZUPT approach. Moreover the quality of the manuscript is overall good. However some major concerns have to be addressed in order to render the manuscript suitable for publication.

Reviewers' comments:

Reviewer's Responses to Questions

**Comments to the Author**

1. Is the manuscript technically sound, and do the data support the conclusions?

Reviewer #1: Yes

Reviewer #2: Yes

2. Has the statistical analysis been performed appropriately and rigorously? 

Reviewer #1: Yes

Reviewer #2: Yes

3. Have the authors made all data underlying the findings in their manuscript fully available?

Reviewer #1: Yes

Reviewer #2: Yes

4. Is the manuscript presented in an intelligible fashion and written in standard English?

Reviewer #1: Yes

Reviewer #2: Yes

5. Review Comments to the Author

Reviewer #1: The paper analyzes the estimation of stride length and cumulative path using foot-mounted inertial sensors during sprinting and adopting a method based on the zero-velocity assumption. Congratulations to the authors, the study is really interesting. In general, the paper has a high-quality English language and the results are clearly presented, but further analyses should be added and discussed.

Introduction

1. This study is focused on the estimation of the stride length/cumulative distance, thus I would suggest to stress in the very first part of the introduction the importance of the estimation of stride length and the foot displacement during running.

2. In line 42 where the zero-velocity assumption is introduced I suggest to cite the following paper: I. Skog, P. Handel, J. -O. Nilsson and J. Rantakokko, "Zero-Velocity Detection—An Algorithm Evaluation," in IEEE Transactions on Biomedical Engineering, vol. 57, no. 11, pp. 2657-2666, Nov. 2010, doi: 10.1109/TBME.2010.2060723.

3. The authors correctly assessed that the displacement estimation with a zero-velocity update (ZUPT) has been highly validated during walking. Examples of accurate ZUPT detectors should be mentioned:

- I. Skog, P. Handel, J. -O. Nilsson and J. Rantakokko, "Zero-Velocity Detection—An Algorithm Evaluation," in IEEE Transactions on Biomedical Engineering, vol. 57, no. 11, pp. 2657-2666, Nov. 2010, doi: 10.1109/TBME.2010.2060723.

- R. Rossanigo, M. Caruso, F. Salis, S. Bertuletti, U. Della Croce and A. Cereatti, "An Optimal Procedure for Stride Length Estimation Using Foot-Mounted Magneto-Inertial Measurement Units," 2021 IEEE International Symposium on Medical Measurements and Applications (MeMeA), Lausanne, Switzerland, 2021, pp. 1-6, doi: 10.1109/MeMeA52024.2021.9478604.

- A. Peruzzi, U. Della Croce and A. Cereatti, "Estimation of stride length in level walking using inertial measurement unit attached to the foot: A validation of the zero velocity assumption", Journal of Biomechanics, vol. 44, pp. 1991-1994, 2011.

4. Please split into two parts the issues of the estimation of foot displacement during sprinting: a) issues related to the hardware (full scale, sample frequency, and various IMU specifications), b) issues related to the inertial data processing (drift, unclear zero-velocity periods).

Please briefly provide the findings of the cited relevant literature in terms of accuracy and validity of ZUPT method(s) always assessing the declared errors.

5. The sampling frequency used in this study seems to be only 1600 Hz in the introduction, please revise this point.

Methods- IMU analysis

1. The authors estimated stride lengths following Potter et al, 2019. Please briefly mention how this method works, in particular the zero-velocity instant selection. How was the stance phase detected? The stationary points are the minima of the angular velocity: did you use the angular velocity magnitude? Did the definition of stride length use all the three foot displacement components?

2. The study by Potter et al, 2019 is cited as [10] but I think it is [12]. Please revise the citations.

Results

1. Please add how many strides per runner you analyzed.

2. The validation was conducted with respect to the stride length/cumulative distance provided by the cameras. Please provide the mean and standard deviation of these reference values.

3. To provide complete results, please add the errors in m and not only percentage errors.

4. The obtained errors were analyzed with respect to the running speed. You calculated the running speed exploiting the calculated distance from IMUs and the duration. I suggest to use the running speed provided by your reference, or at least to clarify the errors in terms of stride speed with respect to the reference values calculated from the cameras.

5. The errors on stride length are only evaluated for the last stride. Why? Please provide the stride-by-stride errors.

6. Figure 2: please use the same range of y-axis.

7. Figure 3: please add ‘∼’ before the reference cumulative distance. Please use the same legend as Figure 2 to distinguish the subjects.

8. Figure 4: this figure shows the average running speed calculated at each ∼5m-distance traveled.

a. Please provide a similar figure but showing a point for each stride speed.

b. To understand the accuracy in the estimation of running speed, add in the same plot the stride speed calculated with the cameras.

c. Is a similar speed pattern during the 70m-sprint shown for each subject? Does the literature confirm that the maximal speed is recorded between 20m and 70m and not at the last stride?

9. Simple linear regression results: please consider to provide a plot showing the stride length errors vs stride speeds.

Discussion

1. Please clarify that you used a zero-velocity detector only based on angular velocities and other detectors can be used to try to improve the results. I suggest again the following detectors to cite, which can be considered also for the comparison of the errors with respect to level walking:

a. I. Skog, P. Handel, J. -O. Nilsson and J. Rantakokko, "Zero-Velocity Detection—An Algorithm Evaluation," in IEEE Transactions on Biomedical Engineering, vol. 57, no. 11, pp. 2657-2666, Nov. 2010, doi: 10.1109/TBME.2010.2060723.

b. R. Rossanigo, M. Caruso, F. Salis, S. Bertuletti, U. D. Croce and A. Cereatti, "An Optimal Procedure for Stride Length Estimation Using Foot-Mounted Magneto-Inertial Measurement Units," 2021 IEEE International Symposium on Medical Measurements and Applications (MeMeA), Lausanne, Switzerland, 2021, pp. 1-6, doi: 10.1109/MeMeA52024.2021.9478604.

2. Please improve the comparison of your errors with respect to the cited relevant literature in terms of stride length and cumulative distance.

3. Please add a conclusive recommendation on the suggested IMU specifications to use. To reach good results, the choice of full scale must be a trade-off between the resolution and the saturation issue.

4. Line 265: the sentence ‘this method … does not require any information about IMU orientation in space’ is misleading. The sensor axes do not have to be aligned with the anatomical ones, but you need to estimate the foot orientation to remove the gravity contribution.

5. Please stress the limitations of this study. For example, the sample frequency of the reference cameras is 240 Hz, thus very lower than the IMU sample frequency: which is the minimal detectable difference in m?

Reviewer #2: This paper describes a study within a longstanding effort to use only shoe-mounted inertial measurement units to track stride characteristics. The study specifically extends the application to high-speed sprinting (6-9+ m/s) and uses higher-performance IMUs in comparison to prior research. The authors do an analysis by the conventional ZUPT method and show that its results appear better than the latest alternative effort that used less high-performance sensors.

This study appears to be well-executed and I have confidence that the algorithm was implemented correctly and the processing done properly. The study is contextualized well within the field and the Methods are reasonably well-described. The results appear reasonable and consistent with the authors' expectations and the predictions of the field, but establishing a proof-in-practice that this system can work for sprinting applications. Overall it is a good study.

I do have a variety of specific comments below that I request the authors to address, to add clarity, improve comprehensiveness, or make adjustments to the interpretation.

Notes on the data set:

1) It contains raw data from all the IMUs, as well as data of tracked stride characteristics from the cameras and stride counts. But it does not include the stride results processed from IMU data, which would be necessary to remake the plots and statistics as reported. I suggest the authors consider another table, similar to the "Kinovea" file, with those results from the IMUs so that readers can further interrogate the results without having to remake the entire process and hope they matched the algorithm perfectly.

2) A brief overview document would be helpful, describing what is in each file and how they relate.

Specific Comments (Line numbers referring to the PDF supplied for review):

lines 54-57: I suggest also adding a mention of sampling rate limitations in this introductory paragraph.

lines 79-84: the last part of this sentence doesn't quite make clear what the authors of [14] did. Can you rework the part starting with "then"?

Table 1: It would be good to include sex, age and if available, estimated maximum speed, for each runner.

Lines 138-140: Using high-g only when the low-g accelerometer saturated: good idea, I was hoping to see this taken into consideration. But, the two accelerometers have different sampling rates, and the high-g accelerometer's rate is not matched to the gyroscope. It would be helpful to the reader if there were a description of how this asynchrony was handled.

Lines 143-145: Detection of stationary periods is a critical, and difficult-to-define, step. Please expand on the details of the customized algorithm for that, or provide a reference (is it the other Potter paper?)

line 151: I was briefly confused by the parenthetical statement here "(i.e., ∼20, ∼50 and !70 meters)", because I thought it meant 20, 50 and 70 were used as the reference values. A re-reading clarified that the real data were used, but the authors may with to adjust phrasing to avoid confusion.

Lines 159-160: Using the IMU-derived stride duration, defined from the stationary periods, can introduce uncertainty if the stationary period is not defined by a "sharp" event. For example, if there is an extended stationary period, then the time when one stride changes to another is not precise. The authors should define the time more precisely than the current text does, perhaps in conjunction with the expansion requested above regarding the detection of the stationary period.

Line 165ff: Please clarify what exactly the raters were looking for. For example: location of the toe during an extended period of ground contact; or, position of the IMU at its lowest point; or... whatever else formally defined the raters; instructions.

line 178: Alpha level was set to 0.05: What statistics did this apply to? Slope of the regression? Bias term? Limits of the Bland-Altman analysis (e.g. 95% = 1-alpha)? Please include that info for the reader.

Lines 182, 195, 186, 187, 190: are the (value +/- number) terms (mean +/- SD)? Please state.

Lines 192-5: I know the results are in Table 3, but it would also help the reader here if you would summarize in words the nature of the effects of speed and body mass. As I read the table, I think it means something like "speed had a negative effect and mass had a positive effect on most forms of error." (but the authors get to straighten that out). See also the comment on Table 3.

Table 3: The Table is highly informative and valuable. But it needs a little more information to help with interpretation, e.g. in the caption or the column headings. Specifically, the units associated with Beta need to be stated: are they [percentage per m/s] and [percentage per kg]? Something else? Above all, I suggest the authors make sure that if a reader looks at any given number in that table, a specific interpretation is unmistakably clear.

Lines 226-8: the final sentence is an overstatement; it's true that the results in this study appear better (lower error) than the results of the de Ruiter study, but with different people and no head-to-head comparison, I'd say this result "suggests" but does not "demonstrate" the superiority of this paper's method/sensor.

Lines 239-240 ff: this section provides indirect evidence, not exactly a "find"-ing (see line 239). As above, it is observational rather than a direct comparison of saturated vs. unsaturated signals. It is all valid commentary, but presentation of the claim should be softened.

Line 250: I'm curious as to why you could not quantify what percentage of data were lost due to saturation. Shouldn't any signal value exactly equal to the maximum reading indicate saturation? If this refers to the later comments in lines 254-8, I suggest moving that commentary up.

Lines 259-262ff: This section argues for practical relevance. I agree that the results look good overall, especially in the context of this field. But in sports where there's about 1.5% difference across an entire championship heat (e.g. times 9.89-10.15 s in NCAA D-1 men's 100 m final), it's unclear what can be learned from a tool with the demonstrated precision. Some additional context and commentary on strengths, uses and limitations would be valuable here. Training perhaps? Perhaps the authors might (optionally) comment on test-retest reliability to support such a thing. Other uses?

Figure 2: The data presented here are very helpful for interpretation. But, the arrangement of the horizontal axis ticks is rather disorienting, since they are sort of between the different subfigures. I suggest moving them onto the zero line within each figure, or perhaps adding a full grid or a vertical-only grid and box outline for each subfigure.

As I look at the data here, the importance of the sensitivity analyses becomes clearer. The authors may wish to expand on the details of the sensitivity to speed. In particular: the conclusion currently presented is that these trends are only a minor effect, and while I don't disagree, the trend as shown does suggest that if used at higher speeds in future testing, the error may become increasingly negative. It's probably worth pointing that out, e.g. in the section where the authors already recommend even higher ranges in future sensors.

Figure 3: I suggest adding to the caption (or elsewhere in the Methods) the specific definition of the limits of agreement (95%?).

6. PLOS authors have the option to publish the peer review history of their article (what does this mean?). If published, this will include your full peer review and any attached files.

Reviewer #1: No

Reviewer #2: No

---

## [Author Response · Author response to Decision Letter 0]

5 Oct 2023

Reviewer #1: The paper analyzes the estimation of stride length and cumulative path using foot-mounted inertial sensors during sprinting and adopting a method based on the zero-velocity assumption. Congratulations to the authors, the study is really interesting. In general, the paper has a high-quality English language and the results are clearly presented, but further analyses should be added and discussed.

Thank you for your comments, we appreciate your positive feedback. We have addressed your specific comments below. Please note that we have added line numbers from the manuscript with the tracked changes and refer to those, while the original feedback was based on the line numbers from the original submission.

Introduction

1. This study is focused on the estimation of the stride length/cumulative distance, thus I would suggest to stress in the very first part of the introduction the importance of the estimation of stride length and the foot displacement during running.

Thank you for your comment. We have added to lines 34-39: “Athletes and coaches looking to improve sprinting performance can assess different sprinting performance determinants, such as stride frequency, stride length and speed during different phases of a sprint and use this information to evaluate races and training runs. Stride length estimates are particularly useful in hurdle and jump events where take off position needs to be optimal.” 

2. In line 42 where the zero-velocity assumption is introduced I suggest to cite the following paper: I. Skog, P. Handel, J. -O. Nilsson and J. Rantakokko, "Zero-Velocity Detection—An Algorithm Evaluation," in IEEE Transactions on Biomedical Engineering, vol. 57, no. 11, pp. 2657-2666, Nov. 2010, doi: 10.1109/TBME.2010.2060723.

Thank you for your suggestion. Please see our response to your 3rd comment on the introduction. 

3. The authors correctly assessed that the displacement estimation with a zero-velocity update (ZUPT) has been highly validated during walking. Examples of accurate ZUPT detectors should be mentioned:

- I. Skog, P. Handel, J. -O. Nilsson and J. Rantakokko, "Zero-Velocity Detection—An Algorithm Evaluation," in IEEE Transactions on Biomedical Engineering, vol. 57, no. 11, pp. 2657-2666, Nov. 2010, doi: 10.1109/TBME.2010.2060723.

- R. Rossanigo, M. Caruso, F. Salis, S. Bertuletti, U. Della Croce and A. Cereatti, "An Optimal Procedure for Stride Length Estimation Using Foot-Mounted Magneto-Inertial Measurement Units," 2021 IEEE International Symposium on Medical Measurements and Applications (MeMeA), Lausanne, Switzerland, 2021, pp. 1-6, doi: 10.1109/MeMeA52024.2021.9478604.

- A. Peruzzi, U. Della Croce and A. Cereatti, "Estimation of stride length in level walking using inertial measurement unit attached to the foot: A validation of the zero velocity assumption", Journal of Biomechanics, vol. 44, pp. 1991-1994, 2011.

Thank you for your suggestion. We have added the suggested references in line 49 and 62, referred as 16, 17 and 18, respectively. 

4. Please split into two parts the issues of the estimation of foot displacement during sprinting: a) issues related to the hardware (full scale, sample frequency, and various IMU specifications), b) issues related to the inertial data processing (drift, unclear zero-velocity periods).

Thank you for your comment. We have now separated this, in lines 59-66: “The implementation of the ZUPT method for sprinting has two main challenges (i.e., issues related to the hardware and issues related to the inertial data processing). The inertial data processing for sprinting is challenging because the foot may not have a clear zero-velocity period, as opposed to level walking [13,16,17,18], which could affect the correction of integration drift errors. The hardware can become an issue because peak accelerations and angular velocities during sprinting might be outside the measurement range of commercially available IMUs and sample frequencies might be insufficient to capture highly dynamic movements, potentially further impacting the external validity of stride lengths obtained with the ZUPT method.”

Please briefly provide the findings of the cited relevant literature in terms of accuracy and validity of ZUPT method(s) always assessing the declared errors.

Thank you for your comment. We have added the errors for reference 19 in lines 68-70 as “Bailey and Harle [19] investigated the accuracy of the traditional ZUPT method for speeds up to 3.4m/s on a treadmill and obtained biases of 0.002±0.029m and 0.03±0.02m/s for the estimation of foot clearance and mean step velocity, respectively” and for reference 21 in lines 70-72 as “Brahms et al. [21] investigated the accuracy of the traditional ZUPT method for speeds up to 4.36m/s during overground running and obtained a bias of -0.032±0.150m for the estimation of stride length”.

5. The sampling frequency used in this study seems to be only 1600 Hz in the introduction, please revise this point.

Thank you for pointing this out. We have modified the sample frequency from 1600Hz to 1125Hz in line 106.

Methods- IMU analysis

1. The authors estimated stride lengths following Potter et al, 2019. Please briefly mention how this method works, inv b particular the zero-velocity instant selection. How was the stance phase detected? The stationary points are the minima of the angular velocity: did you use the angular velocity magnitude? Did the definition of stride length use all the three foot displacement components?

Thank you for your suggestion. We have added a detailed description of how we identified zero velocity periods in lines 160-166: “The detection of stationary periods was done using customized software in Python (Python Software Foundation, Delaware, DE, USA). Firstly, initial contacts were detected as maximum peaks in the resultant acceleration signal. Following Skog et al. [16], thresholds were then adjusted for the gyroscope and acceleration signals to identify 8 samples with the lowest magnitude within the first 25% of the time between initial contacts. These 8 samples represent 5 percent of an average contact time of 150ms that the foot would be stationary. The stationary period was defined as the single sample with the minimum angular velocity during the longest consecutive series of identified samples.”

Yes, we used the three components of foot displacement for the calculation of stride length, following Potter et al. [12].

2. The study by Potter et al, 2019 is cited as [10] but I think it is [12]. Please revise the citations.

Thank you for pointing this out. We have modified the reference from 10 to 12 in line 159.

Results

1. Please add how many strides per runner you analyzed.

Sorry for the confusion. We only analyzed one stride per runner (i.e., the stride at 70m), because the camera view only captured one stride. We have made this clearer by moving the text within this paragraph in lines 128-136: “We placed two cameras (Apple iPhone 12, 1080 pixels at 240Hz), mounted on tripods to capture footfalls at 20m and 70m to establish the exact distance traveled from the start to each footfall closest to each of these marks. Following de Ruiter et al. [14], we placed the iPhone cameras 8 meters away from the track lane that subjects were running in. We choose to place the cameras at 20 and 70m to ensure participants were running near top speed (at 70m) and to separate the acceleration phase from the maximal speed phase (at 20m). The camera at the 70m mark captured the distance between two consecutive footfalls for the right foot to validate the traditional IMU-based ZUPT method for a single stride.”

2. The validation was conducted with respect to the stride length/cumulative distance provided by the cameras. Please provide the mean and standard deviation of these reference values.

Thank you for your suggestion. We have added the mean and standard deviation for the stride length in line 213 as: “and stride lengths from 3.45 to 4.73m (4.07±0.30m)”. We have added the mean and standard deviation for the 20m cumulative distance as: “Participants reached 20-meter distances ranging from 20.25 to 23.44m (22.23±0.99m)”. We have added the mean and standard deviation for the 20-70m cumulative distance as: “Participants ran 20-70-meter ranging from 47.62 to 56.24m (51.32±2.84m)”. We have added the mean and standard deviation for the 70m cumulative distance as: “Participants ran the full 70-meter sprints ranging from 70.39 to 79.24m (73.55±2.55m)”.

3. To provide complete results, please add the errors in m and not only percentage errors.

Thank you for your suggestion. We have added the errors in m for the stride length in line 216 as: “and distance errors from -0.24 to 0.11m with a bias of -0.01±0.19m”. We have added the errors in m for the 20m cumulative distance as “and distance errors from -0.82 to 0.30m and a bias of -0.19±0.76m”. We have added the errors in m for the 20-70m cumulative distance in as: “and distance errors from -2.06 to 0.83m with a bias of -0.62±1.89m”. We have added the errors in m for the 70m cumulative distance as: “and distance errors from -2.72 to 1.04m and a bias of -0.81±2.57m”.

4. The obtained errors were analyzed with respect to the running speed. You calculated the running speed exploiting the calculated distance from IMUs and the duration. I suggest to use the running speed provided by your reference, or at least to clarify the errors in terms of stride speed with respect to the reference values calculated from the cameras.

Thank you for your suggestion. We considered it would be best to use the distance from the cameras and the duration of the IMUs because the IMUs have more temporal resolution than the cameras and the identification of the time of midstance on the camera recordings would not be straightforward.

5. The errors on stride length are only evaluated for the last stride. Why? Please provide the stride-by-stride errors.

Sorry for the confusion. Please see our response to your 1st comment in the results section. We could have analyzed only two strides, one at 20m and another one at 70m. We decided to only analyze the stride at 70m, because compared to the stride at 20m, it is the faster stride.

6. Figure 2: please use the same range of y-axis.

Thank you for your comment. We have modified the y-axis of figure 2 to be in the range -6 to +6.

7. Figure 3: please add ‘∼’ before the reference cumulative distance. Please use the same legend as Figure 2 to distinguish the subjects.

Thank you for your comment. We have used the legend as in Figure 2 to distinguish the subjects.

8. Figure 4: this figure shows the average running speed calculated at each ∼5m-distance traveled.

a. Please provide a similar figure but showing a point for each stride speed.

Thank you for your suggestion. We have updated the figure to clearly show a point for each stride.

b. To understand the accuracy in the estimation of running speed, add in the same plot the stride speed calculated with the cameras.

Please see our response to your 5th comment in the results section. We only could analyze the stride at 70m with the cameras.

c. Is a similar speed pattern during the 70m-sprint shown for each subject? Does the literature confirm that the maximal speed is recorded between 20m and 70m and not at the last stride?

Sorry for the confusion. Participants were asked to do an 80m sprint but we analyzed the stride at 70m because we wanted to ensure that at 70m they were not decelerating for the finish yet. We have added this clarification in lines 24, 27, 29, 103, and throughout. Yes, all our subjects had a similar pattern. Please find below a figure that includes the velocity curves for all participants. Further, the literature confirms that the maximal speed is recorded between 20-70m (Slawinski J, Termoz N, Rabita G, Guilhem G, Dorel S, Morin JB, Samozino P. How 100-m event analyses improve our understanding of world-class men’s and women’s sprint performance. Scand J Med Sci Sports. 2017;27(1):45–54). 

9. Simple linear regression results: please consider to provide a plot showing the stride length errors vs stride speeds.

Please see our response to your 5th comment in the results section. We could only analyze the stride at 70m with the cameras. Other than that, this is in Fig. 2.

Discussion

1. Please clarify that you used a zero-velocity detector only based on angular velocities and other detectors can be used to try to improve the results. I suggest again the following detectors to cite, which can be considered also for the comparison of the errors with respect to level walking:

a. I. Skog, P. Handel, J. -O. Nilsson and J. Rantakokko, "Zero-Velocity Detection—An Algorithm Evaluation," in IEEE Transactions on Biomedical Engineering, vol. 57, no. 11, pp. 2657-2666, Nov. 2010, doi: 10.1109/TBME.2010.2060723.

b. R. Rossanigo, M. Caruso, F. Salis, S. Bertuletti, U. D. Croce and A. Cereatti, "An Optimal Procedure for Stride Length Estimation Using Foot-Mounted Magneto-Inertial Measurement Units," 2021 IEEE International Symposium on Medical Measurements and Applications (MeMeA), Lausanne, Switzerland, 2021, pp. 1-6, doi: 10.1109/MeMeA52024.2021.9478604.

Thank you for your suggestion. Please see our reply to your 1st comment in the methods section. We referenced the study by Skog in line 162.

2. Please improve the comparison of your errors with respect to the cited relevant literature in terms of stride length and cumulative distance.

Thank you for your suggestion. We have improved on the comparison of errors for running and added the modification in lines 258-263 as: “Our findings are similar to those obtained for running at slower speeds (a bias of -0.032 ± 0.150m for overground running up to 4.36m/s [21], but better than those obtained for sprinting (a bias of -2.51±8.54% for peak sprint speeds of 8.42±0.85m/s [14]).”.

3. Please add a conclusive recommendation on the suggested IMU specifications to use. To reach good results, the choice of full scale must be a trade-off between the resolution and the saturation issue.

Thank you for your comment. We have added a conclusive recommendation on the suggested IMU specifications to use in lines 320-327 as: “. Such sensors would not admit gyroscope saturation (>±2400°/s) and have good resolutions (>16bits) and appropriate bandwidths (>500 Hz) and sample frequencies (>1000 Hz) to capture high frequency impacts during ground contact. In commercially available IMUs there are often trade-offs between sensor range, resolution and sampling frequency and we are unaware of any commercial devices that possess all these recommended characteristics. Our data suggests that accurate measures can be obtained with IMUs with specifications similar to the IMU used here (Table 1), but the significant effect of speed also suggests that with the current methods and specifications for even faster speeds stride length is likely to be increasingly under estimated.”.

4. Line 265: the sentence ‘this method … does not require any information about IMU orientation in space’ is misleading. The sensor axes do not have to be aligned with the anatomical ones, but you need to estimate the foot orientation to remove the gravity contribution.

Thank you for pointing this out. We have modified the sentence in lines 313-316 as: “Note that this method only requires a single IMU attached to the athlete’s foot, the IMU does not need to be aligned with anatomical axes and outcomes are minimally affected by subject-specific characteristics (i.e., body mass)”.

5. Please stress the limitations of this study. For example, the sample frequency of the reference cameras is 240 Hz, thus very lower than the IMU sample frequency: which is the minimal detectable difference in m?

For our distance measures, we are not evaluating the positing of for example the sprinter’s head, which indeed is moving ∼40mm between 2 consecutive camera frames (assuming a running speed of 9.6m/s (for simple math)), however we evaluate the foot position during consecutive stance phases. The stance phase is at least 100ms, which means we have about 20 frames where the foot is stationary. As such the camera frame rate does not affect our spatial resolution.

 

Reviewer #2: This paper describes a study within a longstanding effort to use only shoe-mounted inertial measurement units to track stride characteristics. The study specifically extends the application to high-speed sprinting (6-9+ m/s) and uses higher-performance IMUs in comparison to prior research. The authors do an analysis by the conventional ZUPT method and show that its results appear better than the latest alternative effort that used less high-performance sensors.

This study appears to be well-executed and I have confidence that the algorithm was implemented correctly and the processing done properly. The study is contextualized well within the field and the Methods are reasonably well-described. The results appear reasonable and consistent with the authors' expectations and the predictions of the field, but establishing a proof-in-practice that this system can work for sprinting applications. Overall it is a good study.

I do have a variety of specific comments below that I request the authors to address, to add clarity, improve comprehensiveness, or make adjustments to the interpretation.

Thank you for your comments, we appreciate your positive feedback. We have addressed your specific comments below. Please note that we have added line numbers from the manuscript with the tracked changes and refer to those, while the original feedback was based on the line numbers from the original submission.

Notes on the data set:

1) It contains raw data from all the IMUs, as well as data of tracked stride characteristics from the cameras and stride counts. But it does not include the stride results processed from IMU data, which would be necessary to remake the plots and statistics as reported. I suggest the authors consider another table, similar to the "Kinovea" file, with those results from the IMUs so that readers can further interrogate the results without having to remake the entire process and hope they matched the algorithm perfectly.

Thank you for your feedback. We have included a document called “IMU results.xlsx” that contains the results from the IMUs.

2) A brief overview document would be helpful, describing what is in each file and how they relate.

Thank you for this suggestion. We have included a document called “Overview supplemental materials OSF.pdf” that contains a brief overview describing each file attached.

Specific Comments (Line numbers referring to the PDF supplied for review):

lines 54-57: I suggest also adding a mention of sampling rate limitations in this introductory paragraph.

Thank you for your comment. We have added a mention of sampling rate limitations in line 65 as: “and sample frequencies might be insufficient to capture highly dynamic movements”.

lines 79-84: the last part of this sentence doesn't quite make clear what the authors of [14] did. Can you rework the part starting with "then"?

Thank you for your comment. We have reworked the sentence in lines 92-94 as: “, then the velocity offsets were subtracted from the raw velocity signals and all data points prior to the samples with the minimum values were imposed to be zero”.

Table 1: It would be good to include sex, age and if available, estimated maximum speed, for each runner.

We agree that including estimated maximum speed from the IMUs in the table helpful. We have added this information to the table and moved it to the results section (and renumbered the tables). We feel it is less relevant to include sex and age because these will not affect the speed effect on IMU-derived metrics. We did include this information in the methods section: “Seventeen participants over 18 years old were enrolled in this study (4 women, 24.6±6.1yrs, 1.77±0.09m, 71.8±10.3kg, all mean±SD; recruitment period: March 14, 2022 – May 25, 2022).” 

Lines 138-140: Using high-g only when the low-g accelerometer saturated: good idea, I was hoping to see this taken into consideration. But, the two accelerometers have different sampling rates, and the high-g accelerometer's rate is not matched to the gyroscope. It would be helpful to the reader if there were a description of how this asynchrony was handled.

Thank you for pointing this out. We have included this description in lines 150-153 as: “Firstly, the 1600Hz high-g accelerometer signals were linearly down sampled to match the 1125Hz low-g accelerometer signals. Then, a cross-correlation analysis was run to find any phase shift between the two signals. Finally, zero padding was used to make sure the maximum value of the cross correlation occurred at zero-lag”.

Lines 143-145: Detection of stationary periods is a critical, and difficult-to-define, step. Please expand on the details of the customized algorithm for that, or provide a reference (is it the other Potter paper?)

We have added a detailed description of how we identified zero velocity periods in lines 160-166: “The detection of stationary periods was done using customized software in Python (Python Software Foundation, Delaware, DE, USA). Firstly, initial contacts were detected as maximum peaks in the resultant acceleration signal. Following Skog et al. [16], thresholds were then adjusted for the gyroscope and acceleration signals to identify 8 samples with the lowest magnitude within the first 25% of the time between initial contacts. These 8 samples represent 5 percent of an average contact time of 150ms that the foot would be stationary. The stationary period was defined as the single sample with the minimum angular velocity during the longest consecutive series of identified samples.”

line 151: I was briefly confused by the parenthetical statement here "(i.e., ∼20, ∼50 and !70 meters)", because I thought it meant 20, 50 and 70 were used as the reference values. A re-reading clarified that the real data were used, but the authors may with to adjust phrasing to avoid confusion.

Thank you for pointing that out. We have added a clarification in line 173 as “real distance traveled (i.e., ∼20, ∼50 and ∼70 meters)”. 

Lines 159-160: Using the IMU-derived stride duration, defined from the stationary periods, can introduce uncertainty if the stationary period is not defined by a "sharp" event. For example, if there is an extended stationary period, then the time when one stride changes to another is not precise. The authors should define the time more precisely than the current text does, perhaps in conjunction with the expansion requested above regarding the detection of the stationary period.

In addition to the description added in relation to your comment above, we have added a clarification in line 175 “(i.e., the time between two consecutive stationary periods) “. 

Line 165ff: Please clarify what exactly the raters were looking for. For example: location of the toe during an extended period of ground contact; or, position of the IMU at its lowest point; or... whatever else formally defined the raters; instructions.

We have added a more detailed explanation on exactly what the raters were looking for in lines 187-191 as: “For Struth, we identified video frames at or near midstance for two consecutive right steps. Then we placed a marker on top of the IMU and drew the vertical projection of the IMU to the ground. We were then able to calculate the horizontal distance between the two vertical projections to obtain stride length. For Dtruth, we calculated the horizontal distance between the vertical projection of the IMU to the tape mark, denoting 20 or 70 meters from the start.”.

line 178: Alpha level was set to 0.05: What statistics did this apply to? Slope of the regression? Bias term? Limits of the Bland-Altman analysis (e.g. 95% = 1-alpha)? Please include that info for the reader.

Thank you for pointing this out. We have added to what statistics did the alpha level apply to in lines 208-209 as: “Alpha level was set a priori to 0.05 for the slope of the regression”.

Lines 182, 195, 186, 187, 190: are the (value +/- number) terms (mean +/- SD)? Please state.

Yes, we have added this clarification upon first use in line 112 and in the methods under statistics.

Lines 192-5: I know the results are in Table 3, but it would also help the reader here if you would summarize in words the nature of the effects of speed and body mass. As I read the table, I think it means something like "speed had a negative effect and mass had a positive effect on most forms of error." (but the authors get to straighten that out). See also the comment on Table 3.

Thank you for your suggestion. We have included a description of the effects in lines 228-232 as: “Speed had a large significantly negative effect and body mass had a small significantly positive effect on the individual IMU-based stride length error, and on the 20-70-meter cumulative distance error. Speed had a large significantly negative effect on the 70-meter cumulative distance error”.

In the methods, in lines 206-208 we have added a description of the effects as: “The relative magnitude of the effects of different variables were quantified with the standardized betas (β), with β<0.29 being a small effect, 0.30<β<0.49 being a medium effect, β>0.50 being a large effect”.

Table 3: The Table is highly informative and valuable. But it needs a little more information to help with interpretation, e.g. in the caption or the column headings. Specifically, the units associated with Beta need to be stated: are they [percentage per m/s] and [percentage per kg]? Something else? Above all, I suggest the authors make sure that if a reader looks at any given number in that table, a specific interpretation is unmistakably clear.

Thank you for pointing this out. We have included the units of the beta’s in the table as (% per m/s) and (% per kg).

Lines 226-8: the final sentence is an overstatement; it's true that the results in this study appear better (lower error) than the results of the de Ruiter study, but with different people and no head-to-head comparison, I'd say this result "suggests" but does not "demonstrate" the superiority of this paper's method/sensor.

Thank you for your comment. We changed “demonstrates” to “suggests” in line 266.

Lines 239-240 ff: this section provides indirect evidence, not exactly a "find"-ing (see line 239). As above, it is observational rather than a direct comparison of saturated vs. unsaturated signals. It is all valid commentary, but presentation of the claim should be softened.

Yes, we agree. We changed “we found” to “our findings suggest” in line 279.

Line 250: I'm curious as to why you could not quantify what percentage of data were lost due to saturation. Shouldn't any signal value exactly equal to the maximum reading indicate saturation? If this refers to the later comments in lines 254-8, I suggest moving that commentary up.

Yes, that is correct, any signal value equal to the maximum reading indicates saturation. We saw that for every stance phase about 3-4 samples exhibited saturation, but we could not quantify exactly the amount of signal that was lost, as opposed to Potter et al. [12] where the authors artificially saturated the signals and could therefore quantify the amount of signal that was lost.

Lines 259-262ff: This section argues for practical relevance. I agree that the results look good overall, especially in the context of this field. But in sports where there's about 1.5% difference across an entire championship heat (e.g. times 9.89-10.15 s in NCAA D-1 men's 100 m final), it's unclear what can be learned from a tool with the demonstrated precision. Some additional context and commentary on strengths, uses and limitations would be valuable here. Training perhaps? Perhaps the authors might (optionally) comment on test-retest reliability to support such a thing. Other uses?

Thank you for your suggestion. We have elaborated on the practical relevance in relation to improving sprinting performance in lines 300-311 as: “While the bias for stride length that we obtained was only -0.27%, the 95% limits of agreement were ±4.6%. In elite sports, 1% differences in performance could be the difference between a gold medal and no medal. Without the presence of IMU saturation, the accuracy of estimated stride lengths during sprinting could be improved and can be of practical relevance in training applications. Stride length estimates are particularly useful in hurdle and jump events where take off position needs to be optimal. Further, accurate stride length estimations allow for accurate speed calculation. From the sprint speed curve presented in Fig 4, sprinting performance determinants such as maximal speed [1] can be extracted. Such a speed curve could also be supplemented with the method proposed by Samozino et al. [24] to obtain estimates of force outputs (e.g., ratio of force). Additionally, the estimates of sensor displacement and orientation used in the ZUPT method can be used to obtain other stride parameters that could be levered in sprint training settings (e.g., contact time, swing time, step frequency, plantar- and dorsiflexion angles).”

Figure 2: The data presented here are very helpful for interpretation. But, the arrangement of the horizontal axis ticks is rather disorienting, since they are sort of between the different subfigures. I suggest moving them onto the zero line within each figure, or perhaps adding a full grid or a vertical-only grid and box outline for each subfigure.

Thank you for your suggestion. We have added a vertical grid to Figure 2.

As I look at the data here, the importance of the sensitivity analyses becomes clearer. The authors may wish to expand on the details of the sensitivity to speed. In particular: the conclusion currently presented is that these trends are only a minor effect, and while I don't disagree, the trend as shown does suggest that if used at higher speeds in future testing, the error may become increasingly negative. It's probably worth pointing that out, e.g. in the section where the authors already recommend even higher ranges in future sensors.

Thank you for your comment. We have pointed that out in line 324-327 as: “Our data suggests that accurate measures can be obtained with IMUs with specifications similar to the IMU used here (Table 1), but the significant effect of speed also suggests that with the current methods and specifications for even faster speeds stride length is likely to be increasingly under estimated”.

Figure 3: I suggest adding to the caption (or elsewhere in the Methods) the specific definition of the limits of agreement (95%?).

Thank you for your suggestion. We have added this in the methods and in the caption as: “Limits of agreement are defined as bias ± 1.96xSD (i.e., 95%).”.

---

## [Decision Letter · Decision Letter 1]

15 Jan 2024

Assessing the validity of the zero-velocity update method for sprinting speeds

PONE-D-23-19191R1

Dear Dr. Hoogkamer,

We’re pleased to inform you that your manuscript has been judged scientifically suitable for publication and will be formally accepted for publication once it meets all outstanding technical requirements.

Kind regards,

Andrea Tigrini, Ph.D.

Academic Editor

PLOS ONE

Additional Editor Comments (optional):

The paper was consistently updated and deserves to be pubished.

Reviewers' comments:

Reviewer's Responses to Questions

**Comments to the Author**

1. If the authors have adequately addressed your comments raised in a previous round of review and you feel that this manuscript is now acceptable for publication, you may indicate that here to bypass the “Comments to the Author” section, enter your conflict of interest statement in the “Confidential to Editor” section, and submit your "Accept" recommendation.

Reviewer #1: All comments have been addressed

Reviewer #2: All comments have been addressed

2. Is the manuscript technically sound, and do the data support the conclusions?

Reviewer #1: Yes

Reviewer #2: Yes

3. Has the statistical analysis been performed appropriately and rigorously? 

Reviewer #1: Yes

Reviewer #2: Yes

4. Have the authors made all data underlying the findings in their manuscript fully available?

Reviewer #1: Yes

Reviewer #2: Yes

5. Is the manuscript presented in an intelligible fashion and written in standard English?

Reviewer #1: Yes

Reviewer #2: Yes

6. Review Comments to the Author

Reviewer #1: (No Response)

Reviewer #2: Thank you for addressing the comments.

I have only one new comment, which relates to the Limits of Agreement analysis in Figure 2. A typical LOA plots the Difference between two measurement methods on the vertical axis, vs. the Mean of those two methods on the horizontal axis. In the figure as presented, it is Difference vs. Reference. I can see reasons for this decision and for my part will leave it to the authors' discretion if that is the form they want to present; but I just wanted to point out that it is nonstandard.

7. PLOS authors have the option to publish the peer review history of their article (what does this mean?). If published, this will include your full peer review and any attached files.

Reviewer #1: No

Reviewer #2: No

---

## [Editor Report · Acceptance letter]

30 Jan 2024

PONE-D-23-19191R1 

PLOS ONE

Dear Dr. Hoogkamer, 

I'm pleased to inform you that your manuscript has been deemed suitable for publication in PLOS ONE. Congratulations! Your manuscript is now being handed over to our production team.

Kind regards, 

on behalf of

Dr. Andrea Tigrini 

Academic Editor

PLOS ONE